# Effects of Frequency Filtering on Intensity and Noise in Accelerometer-Based Physical Activity Measurements

**DOI:** 10.3390/s19092186

**Published:** 2019-05-11

**Authors:** Jonatan Fridolfsson, Mats Börjesson, Christoph Buck, Örjan Ekblom, Elin Ekblom-Bak, Monica Hunsberger, Lauren Lissner, Daniel Arvidsson

**Affiliations:** 1Center for Health and Performance, Department of Food and Nutrition, and Sport Science, University of Gothenburg, SE-40530 Gothenburg, Sweden; mats.brjesson@telia.com (M.B.); daniel.arvidsson@gu.se (D.A.); 2Institute of Neuroscience and Psychology, University of Gothenburg, 40530 Gothenburg, Sweden; 3Department of Medicine and Geriatrics, Sahlgrenska University Hospital/Östra, SE-416 50 Gothenburg, Sweden; 4Department of Biometry and Data Management, Leibniz Institute for Prevention Research and epidemiology (BIPS), DE-283 59 Bremen, Germany; buck@leibniz-bips.de; 5Åstrand Laboratory of Work Physiology, The Swedish School of Sport and Health Sciences, SE-114 86 Stockholm, Sweden; orjan.ekblom@gih.se (Ö.E.); elin.ekblombak@gih.se (E.E.-B.); 6Department of Public Health and Community Medicine, Institute of Medicine, Sahlgrenska Academy, University of Gothenburg, SE-413 90 Gothenburg, Sweden; monica.hunsberger@gu.se (M.H.); lauren.lissner@gu.se (L.L.)

**Keywords:** calibration, ActiGraph, Axivity, children, adults, I.Family, LIV-2013

## Abstract

In objective physical activity (PA) measurements, applying wider frequency filters than the most commonly used ActiGraph (AG) filter may be beneficial when processing accelerometry data. However, the vulnerability of wider filters to noise has not been investigated previously. This study explored the effect of wider frequency filters on measurements of PA, sedentary behavior (SED), and capturing of noise. Apart from the standard AG band-pass filter (0.29–1.63 Hz), modified filters with low-pass component cutoffs at 4 Hz, 10 Hz, or removed were analyzed. Calibrations against energy expenditure were performed with lab data from children and adults to generate filter-specific intensity cut-points. Free-living accelerometer data from children and adults were processed using the different filters and intensity cut-points. There was a contribution of acceleration related to PA at frequencies up to 10 Hz. The contribution was more pronounced at moderate and vigorous PA levels, although additional acceleration also occurred at SED. The classification discrepancy between AG and the wider filters was small at SED (1–2%) but very large at the highest intensities (>90%). The present study suggests an optimal low-pass frequency filter with a cutoff at 10 Hz to include all acceleration relevant to PA with minimal effect of noise.

## 1. Introduction

Objective measurement of physical activity (PA) has become widespread during the last decade and has provided researchers with more detailed data compared to self-report methods. With this, stronger relationships between PA and various health aspects have been established [1]. PA measurement has become standard in large scale epidemiological research such as the National Health and Nutrition Examination Survey (NHANES) and the UK Biobank study with 15,000–100,000 participants [2,3]. Accelerometers are the most commonly used objective methods and can be worn on different body locations (waist, wrist, or thigh) for multiple days, where the waist has been the most widely used location. After data collection, recorded raw acceleration data has to be processed to useful PA metrics. Most studies have processed raw data to the output ActiGraph counts [4]. ActiGraph (AG) is the most commonly used accelerometer in PA research [5], and the output from the first AG accelerometers was only presented in counts. Although raw acceleration is accessible from newer AG accelerometers, counts is still the most used metric [4].

To generate AG counts, the acceleration is filtered, rectified, and aggregated over a time window called epoch that is usually 60 s long. The processing to AG counts has previously only been possible with AG accelerometers but the details of replicating this process are now available enabling creation of AG counts from raw data collected by any sensor [6]. After raw acceleration data have been processed to a useful movement intensity metric a calibration against PA is performed, normally using energy consumption expressed as metabolic equivalents (METs) [4]. However, when translating the AG counts to represent PA some limitations become apparent. First, at higher intensities of PA equivalent to running, the counts reach a plateau [7]. Second, adults generate more counts than children during the same locomotion speed, whereas the opposite occurs with other processing [8,9,10]. These limitations are explained biomechanically by the narrow frequency filter involved in the processing of AG counts removing the influence of different step frequency [11]. It is known, that the step frequency is higher in children than adults and during more intensive PA in both adults and children [12,13], which leads to more acceleration being filtered out. Widening the frequency filter improves the measurement of high-intensity PA and better captures variations in movement pattern [11]. 

Although it has been demonstrated that a wider frequency filter improves measurement of PA intensity in a lab setting, it might be vulnerable to inclusion of more noise in a free-living setting [14]. Most processing methods apart from AG do not involve any kind of frequency filter to remove potential noise [15,16] but the effect of the absence of a filter on the amount of noise captured has not been investigated. It is hypothesized that since the noise is expected to be independent of PA intensity, the relative contribution of noise with wider filters might be higher at low PA intensity potentially classifying more sedentary time as light activity. In addition, it is also expected that there is a larger classification discrepancy between wider filters and the AG filter at high-intensity PA.

The aim of the current study is to explore the optimal low-pass frequency filter that captures most of the acceleration signal relevant to determine PA intensity with minimal effect of noise in lab data and a free-living sample of children and adults.

## 2. Materials and Methods

### 2.1. Study Design

The study consists of two parts:A calibration part in a lab setting with children and adults walking and running on a treadmill at different speeds while measuring acceleration and oxygen consumption. The results of this part enabled assessment of agreement between the outputs from the different frequency filters for the free-living part.A free-living part with children and adults where the effect of different frequency filters applied to accelerometer data was explored. We anticipated that the wider filters would contribute to disproportionally more acceleration at higher intensities. Furthermore, noise would have larger effect at the sedentary level and contribute to misclassification of sedentary to light activity.

### 2.2. Study Sample

The study sample for the calibration part consisted of 10 adults (5 females) mean age 29.2 (SD 6.6) and 10 children (3 females) mean age 10.1 (SD 0.6). The recruitment of this sample has been described elsewhere [11]. For the free-living part of the study, data from the LIV-2013 study [17] and the Swedish sample of the second follow-up of the I.Family study conducted in 2013–2014 [18] were analyzed. The LIV-2013 sample consisted of 375 adults (24% female) mean age 48.4 (SD 11.6) and the I.Family sample consisted of 303 children (50% female) mean age 10.9 (SD 2.4). More detailed descriptions of the recruitment and characteristics of these samples are available elsewhere [17,18]. 

### 2.3. Data Collection

Subjects in the calibration part wore an accelerometer, Axivity AX3 (Axivity Ltd., Newcastle upon Tyne, UK), over their right hip attached to an elastic band around their waist. Subjects were resting, walking, and running on a treadmill while their oxygen consumption (VO_2_) was measured using the stationary metabolic system Oxycon Pro (Jaeger, BD Corporation, Franklin Lakes, NJ, USA). A face mask with a turbine flow meter and expired air sampler was used and VO_2_ was calculated breath by breath. Subjects were instructed to fast for the four hours prior to the experiment and to avoid strenuous exercise the same day. The test started with seated rest in an armchair for 20 min in order to reach resting metabolic rate (RMR) [19]. The treadmill protocol consisted of 4 min at each speed of walking at 3, 4, 5, and 6 km∙h^−1^ and running at 8 and 10 km∙h^−1^. The experimental setup has been described previously [11]. The sample rate and acceleration range of the accelerometers were set to 100 Hz and ±8 g, respectively, and the data was extracted with the OmGUI software (Axivity Ltd., Newcastle upon Tyne, UK).

The data from both free-living samples were collected with ActiGraph GT3X+ accelerometers worn over the right hip in an elastic belt around the waist. The subjects were instructed to wear the accelerometer for seven days while keeping their normal PA habits and to remove it when sleeping and during water-based activities. The accelerometers were set to record with a sample rate of 30 Hz and an acceleration range of ±6 g (where 1 g is equivalent to Earth’s gravity), and idle sleep mode was enabled. The raw triaxial acceleration data were extracted using the freely available software actigraph_gt3x_extract [20].

### 2.4. Data Analysis

The calibration data was down-sampled to 30 Hz and original AG counts were generated, as well as modified counts using altered frequency filters. The output from the three axes were then combined to a vector magnitude. The original ActiGraph band-pass filter has high-pass and low-pass half-power cutoff frequencies at 0.29 Hz and 1.63 Hz, respectively. The modified filters were implemented as digital fourth-order Butterworth band-pass filters with the same high-pass threshold as the original ActiGraph filter but with an altered low-pass cutoff either set to 4 or 10 Hz or completely removed with only the high-pass (HP) component remaining. The upper limit of the normal human step frequency is 4 Hz and the majority of the acceleration is therefore below this cutoff [11,12]. However, previous studies have shown that acceleration related to physical activity is found at frequencies up to 10 Hz [11,21]. The steps of converting raw acceleration to counts are shown in Figure 1 and described more in detail by Brønd et al. [6]. When using the modified filters, no previous antialiasing filter was applied nor was the data down-sampled to 10 Hz as with the original method. It has been suggested to move away from the count metric to achieve more transparent methods of measuring physical activity [22]. Therefore, the acceleration output was presented both as mean g×10^−3^ (mg) and aggregated counts. One count is equivalent to 16.64 mg because of the conversion of the original AG range ± 2.13 g → 2 × 2.13 g × 10^3^ = 4260 mg (r) to 8-bit resolution 2^8^ = 256 (b) [23]. In order to present mean acceleration instead of aggregated, the counts have to be divided by the sample rate prior to aggregation, which is 10 Hz (f), and the epoch length (e). Original AG counts with 60 s epochs will therefore be converted to mg by multiplying with 0.0277 as shown in Equation (1): (1)$$\frac{r}{b\times f\times e}=\frac{4260\;\mathrm{mg}}{256\times 10\;\mathrm{Hz}\times 60\;s}=0.0277\;\mathrm{mg}$$

However, because of the 8-bit resolution, only integer counts are aggregated whereas decimals are also considered when calculating the mean mg of the acceleration. Therefore, the different output presented in this study will not perfectly agree by a factor of 0.0277. Since the AX3 accelerometers used an acceleration range of ±8 g instead of ±6 g with the GT3x, the calibration data was truncated to ±6 with the modified counts. 

VO_2_ data from the 20 min of resting was filtered with a moving average filter with a window size of 2 min and the minimum value was considered the individual RMR. One minute of data captured beginning at 2 min and 45 s of duration at each treadmill speed (i.e., 2:45–3:45) was used for calibration of acceleration output against energy expenditure. VO_2_ data captured during this minute was averaged and converted to METs by dividing with the individual RMR. Acceleration captured during the same minute was processed to mean filtered acceleration in mg as well as converted to counts and aggregated to a one-minute epoch. Separate smoothing splines for children and adults were fitted to the METs and acceleration output from all locomotion speeds using MATLAB ‘fit’ function with a smoothing factor of 0.1 and a forced starting point of zero acceleration and one MET. MET cut-points at <1.5, ≥1.5–<3, ≥3–<6, ≥6–<9, ≥9 were implemented to represent sedentary (SED), light (LPA), moderate (MPA), vigorous (VPA), and very vigorous PA (VVPA) respectively, which is in line with previous literature [24]. The smoothing splines generated from the output from each processing method were evaluated for the MET cut-points generating filter- and age-specific cut-points in mg and counts.

Free-living accelerometer data were processed with the original AG filter and with modified filters as previously described using epoch lengths of 1, 3, 10, and 60 s. Shorter epoch lengths better capture the extremes of physical behavior which leads to higher amounts of SED and VPA but less LPA and MPA [25]. Children’s physical activity is usually more intermittent and would therefore be better captured with shorter epoch lengths [26]. However, the half-power frequency of the high-pass component of the AG filter of 0.29 Hz allows a signal related to slow movement with a cycle of approximately 3 s to pass [6]. The associations between 1 s epochs and health variables are stronger when using bouts of 5 s [25], which would also support that a slightly longer epoch length is favorable. Therefore, the main results of this study were based on three-second epochs although 1, 10, and 60 s epochs were also analyzed. 

Only valid days with at least 12 h of wear time were included in the analysis. Samples in which the sensor status was idle according to the idle sleep-mode were considered non-wear time. More subjects in the adult group wore the accelerometer during sleep compared to the children; therefore, the acceleration recorded between 23.00–06.00 was not included in the analysis. 

Acceleration outputs with different filters were classified according to the cut-points retrieved by the calibration in the current study. The classification of the modified-filters output was then compared epoch by epoch to the AG output. 

For further analysis of the frequency content in free-living measurements, the data were analyzed using a set of frequency sub-bands. Apart from the original AG filter with a pass band of 0.29–1.63 Hz, three additional sub-bands were investigated using tenth-order Butterworth filters; band-pass with half-power frequencies at (1) 1.7–4 Hz and (2) 4–10 Hz as well as (3) a high-pass filter with a half-power frequency at 10 Hz. The dispersion of the absolute amount of acceleration recorded with regard to intensity was analyzed for each of the four sub-bands. Three-second AG-filtered samples from each subject were divided into 41 equally spaced bins, from 0–10 mg to above 400 mg, using the MATLAB ‘histcounts’ function. The three-second samples from each of the four sub-bands was then aggregated for each bin-index from the dispersion of the AG output, thus generating a histogram of the acceleration content of the sub-bands by aggregating the samples with regard to the intensity of the AG output. Data analysis was performed in MATLAB 2018b (MathWorks, Natick, MA, USA). 

## 3. Results

### 3.1. Calibration

Cut-points generated from the calibration of acceleration output against energy expenditure in mg and counts∙min^−1^ for MET values of 1.5, 3, 6, and 9 are shown in Table 1. In Figure 2, acceleration output is plotted against METs for all locomotion speeds and processing methods together with the fitted smoothing splines that were evaluated to generate the cut-points. The figure demonstrates a substantial increase of acceleration at higher intensity when widening the filter. 

### 3.2. Free-living

The mean (SD) amount of data per subject in the free-living part of the study was 45.1 (31.4) hours in the adult group and 65.5 (43.1) hours in the child group only including data within 06.00–23.00 and from valid days of at least 12 h. 

Figure 3a shows the absolute amount of acceleration aggregated in three-second epochs generated by each frequency sub-band filter. The intensity of the AG epoch was considered as reference on the x-axis. The peaks of the sub-bands indicate at what intensity most of the acceleration was generated. Both children and adults generated most of the acceleration as MPA. In adults, the peaks of the higher frequency sub-bands are slightly shifted toward higher intensities, whereas they are at approximately the same intensity among children. In Figure 3b, the aggregated acceleration from the higher-frequency bands is expressed relative to AG. The higher frequency content is not distributed evenly; instead, the contribution is dependent on intensity. The relative contributions from the higher-frequency sub-bands are greater at moderate to vigorous intensity (MVPA). However, the peaks of the relative contribution among children are at MPA, whereas these peaks are shifted to VVPA among adults. There is also an additional contribution of counts at the lowest intensity equivalent to SED.

Using the cut-points retrieved from the calibration part of this study, the classification agreement between AG and the modified filters is shown as confusion charts in Figure 4. There is a small misclassification at about 1–2% of SED among the wider filters compared to AG. The effect of the wider filters on the classification of moderate and more intense activity is very large with an agreement of 30–80% at MPA and about 10% or less at higher intensities. All investigated epoch lengths generated similar classification patterns. Confusion charts generated with epoch lengths of 1, 10, and 60 s are available as Appendix A. To clarify the effect on the actual amount of PA captured by the different filters in a free-living setting, the distribution between intensity levels can be seen in Table 2. There are slightly more epochs classified as SED or LPA with the wider filters, whereas the amount of MVPA is considerably less. 

## 4. Discussion

The results show that the higher-frequency sub-bands contain acceleration primarily related to PA of moderate intensity and higher. The effect of wider filters with regard to more noise being captured during low-intensity physical behavior is very small since only 1–2% of SED is classified as LPA, whereas the wider filters have a major impact on the measurement of PA of higher intensity with 88–96% disagreement at these intensities.

The relationship between METs and accelerometer output is clearly curvilinear as seen in Figure 2. The shape of this relationship is also different between the filters where AG is more linear at low intensities. In the current study, when fitting a line to represent this relationship it is of great importance to keep the curvilinear shape to allow comparisons between the filters, especially between 1 and 3 METs where the low-intensity cut-points are found. Previous calibration studies have primarily been using linear regression or receiver operating characteristic (ROC) curves to identify cut-points [27], although there are examples of calibrations retrieved from smoothing splines [28]. ROC curves have been suggested to be more accurate than regression models [29], but the use of ROC curves requires a dataset with continuous data points across the intended cut-points which was not the case in this study. However, the cut-points generated from this calibration (Table 1) are similar to previous calibrations of AG vector magnitude counts [4]. All fitted smoothing splines reach a plateau after 10 METs, which makes them unsuitable to use for higher intensities than this. This is probably caused by the low number of subjects reaching these intensities and therefore the fitted line is highly influenced by single subjects. However, the models should still be sufficient to establish the VVPA cut-point set at 9 METs. Interestingly, the fitted lines change place with wider filters, from children generating more acceleration at a particular MET level to adults generating more acceleration. This is expected because with the wider filters, children and adults generate the same amount of acceleration [11], but children have lower MET values due to their higher RMR [30]. 

The larger relative contribution from the higher-frequency sub-bands at higher intensities (Figure 3b) is expected because of the higher step frequency associated with higher intensities [11]. Still, these results are important as they show that the higher-frequency bands do not provide the same acceleration information as when using the AG filter, but instead are specifically associated with higher intensities. Flat horizontal lines of the relative contribution would have indicated no meaningful addition to the measurement. 

With the higher-frequency sub-bands, the peaks of the relative contribution is shifted to higher intensities among adults compared to children. The intensity of the adult peaks, VVPA, is equivalent to running. Running is a continuous activity with high step frequency around 3 Hz [12], that clearly would be better captured by a wider frequency filter. Children rarely perform continuous PA such as running [26], but still perform far more VVPA than adults (Table 2). This suggests that children generate their high-intensity PA by other, more varied and possibly intermittent, activities, which could explain the different relative high-frequency contribution between age groups. However, the amount of VPA and higher among adults is very low, which leads to large relative differences although the absolute differences between the frequency bands are small.

The difference in classification of the higher intensities between AG and the wider filters are remarkably high with a classification agreement of about 10% or lower at VPA and VVPA (Figure 4). The AG filter hinders classification of higher intensity by not accounting for variations in step frequency [11]. With higher intensities, the epochs are therefore more or less classified by chance with AG. The agreement is less among children, which is expected because of their higher step frequency [12,13]. 

There was also a relative contribution of acceleration gathered at the lowest intensity equivalent to SED (Figure 3b). Additional contribution of counts at this intensity might indicate more noise being captured in a free-living setting. On the other hand, Figure 4 suggests that this is not a major concern. According to the confusion chart, only 1–2% of the epochs classified as SED by AG were classified as LPA by the modified filters. Because of differences in signal properties to which the wider filters are sensitive, it is inevitable that there is some variation in the classification. Epochs below but close to the AG LPA cut-point might just reach above the LPA cut-point with the wider filters. The total amount of SED does not decrease with the wider filters but rather increases slightly. Even if a wider frequency filter would capture more acceleration considered as noise than a narrow one, this might not lead to more misclassification of PA. This is because more absolute acceleration has to be gathered with wider filters compared to narrow ones to reach a certain cut-point as seen in Table 1. However, this is only the case if acceleration related to physical activity is expected in the whole frequency pass-band. 

An optimal filter should capture all acceleration related to PA but not acceleration considered to be noise [14]. Acceleration related to human locomotion is found between 1 and 10 Hz [21]. Widening the filter further than 10 Hz will not capture more acceleration related to locomotion, which is why the difference between 10 Hz and HP cut-points is small (Table 1). The frequency content of potential noise has not been thoroughly investigated but it has been shown that acceleration related to car driving is at frequencies above 10 Hz [31]. Although the HP filter investigated in the current study did not misclassify more SED epochs on group level, it might do so in individuals spending a lot of time in cars. Therefore, we suggest an optimal low-pass cutoff in frequency filtering of accelerometry data of 10 Hz. 

Physical activity recommendations are particularly aimed toward MVPA because the high-intensity physical activity seems to be of most importance for health [32]. Considering that wider filters previously have been shown to improve the measurement of MVPA [11], the classification discrepancy in the present study suggests that AG seriously misclassifies MVPA in a free-living setting. In total, the amount of MVPA captured is less with wider filters (Table 2). It might be assumed that wider frequency filters that better capture MVPA would capture more of this in free-living measurements. However, AG rather seems to overestimate some epochs. These results are in line with previous research where less MVPA has been captured using an unfiltered acceleration metric compared to AG [33]. Although less MVPA was captured, the amount captured seems to reflect the actual PA better. 

An unfiltered acceleration metric has also been shown to have a higher association with energy expenditure measured by doubly labeled water [34]. More accurate capturing of the individual physical activity level could have major implications for measurement of PA and its relation to health. The results of the current study suggest that it is not possible to distinguish between MPA, VPA, and VVPA with the AG filter. A 10 Hz filter enables distinguishing of these intensities without being more vulnerable to capturing noise. Although wider filters capture less high-intensity PA, the amount captured might be more precise and possibly show stronger relationships toward health variables. Future studies should therefore investigate the relationship between the 10 Hz filtered output and health variables compared to the standard AG-filtered output. 

## 5. Conclusions

The present study suggests that a low-pass frequency filter with a cut-off at 10 Hz might capture most of the relevant acceleration signals to be able to determine PA intensity with minimal effect of noise under free-living in children and adults. These conclusions are based on the observations of no additional relevant information above 10 Hz, an important contribution of acceleration to the MVPA level with the 10 Hz filter and with only small misclassification of sedentary into light activity. By enabling improvements in the current PA methodology, these findings can potentially improve investigations and current understanding of the relationship between high-intensity PA and health.

## Figures and Tables

**Figure 1 sensors-19-02186-f001:**
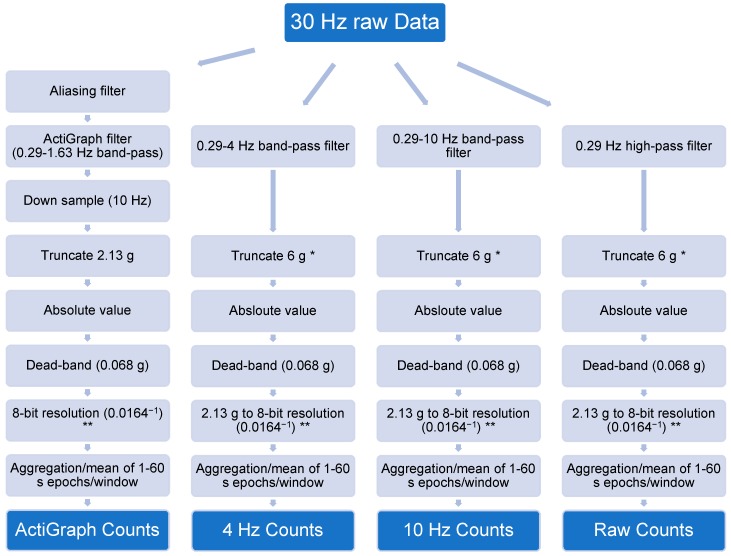
Description of the original ActiGraph and the modified processing methods. * Truncation to 6 g with the modified methods was only performed with the calibration data. ** Converting acceleration in g to counts with a 2.13 g to 8-bit resolution was only performed with the output presented in counts.

**Figure 2 sensors-19-02186-f002:**
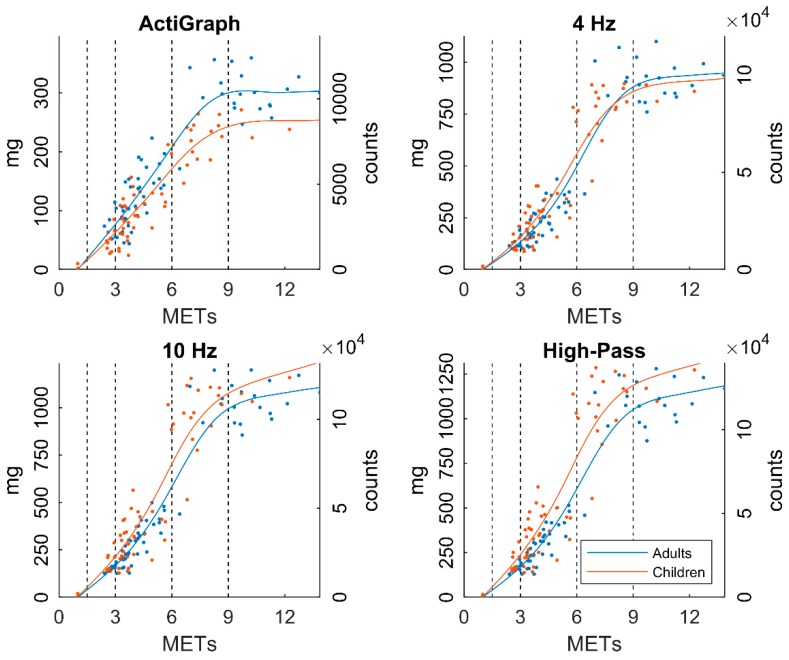
MET values plotted against acceleration output (mg, counts) using different filters for all locomotion speeds and subjects as well as fitted smoothing splines. Dotted lines represent cut-points at 1.5 (LPA), 3 (MPA), 6 (VPA), and 9 (VVPA) METs, respectively.

**Figure 3 sensors-19-02186-f003:**
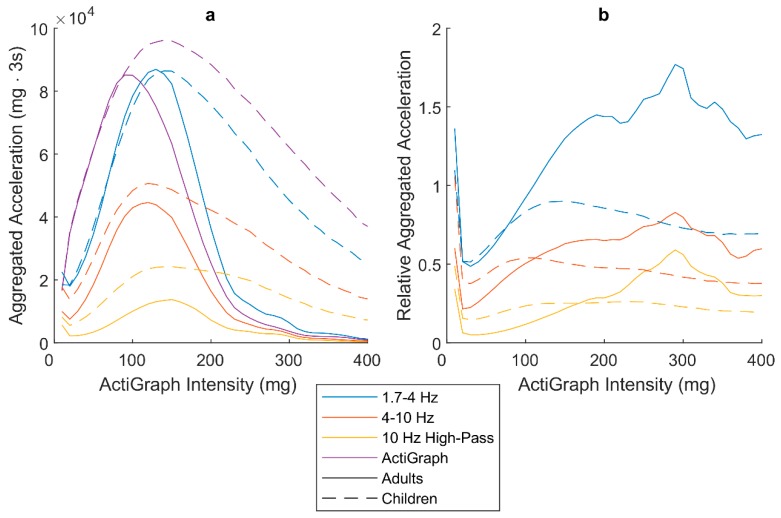
(**a**) Absolute aggregated three-second samples of acceleration from the sub-bands. (**b**) Aggregated acceleration relative to the ActiGraph filter.

**Figure 4 sensors-19-02186-f004:**
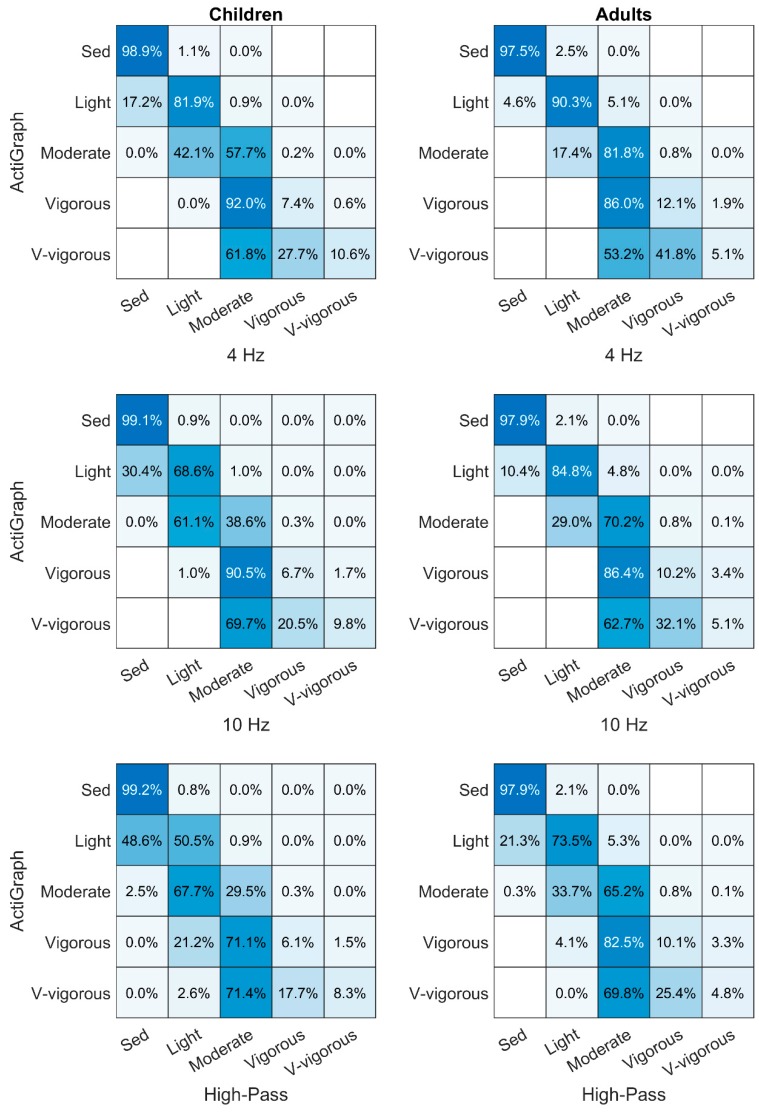
Row-normalized confusion charts comparing ActiGraph-filtered output with output from the modified filters epoch by epoch. Numbers on the diagonal from the upper left to lower right corner with 100% would indicate a perfect agreement between the filters whereas numbers in the lower or upper triangles from the diagonal indicate that the modified filters classify the activity intensity as lower or higher, respectively.

**Table 1 sensors-19-02186-t001:** Cut-points for acceleration output in mg (counts∙min^−1^) retrieved by evaluating smoothing splines fitted to the acceleration output and energy expenditure for each age group.

Filter	Age Group	1.5 METs (LPA)	3 METs (MPA)	6 METs (VPA)	9 METs (VVPA)
**Children**	ActiGraph	16.1 (531)	63.0 (2100)	171.0 (5865)	242.9 (8417)
	4 Hz	37.4 (3798)	158.5 (16,326)	557.9 (59,164)	858.8 (91,721)
	10 Hz	51.9 (5345)	214.1 (22,275)	703.5 (74,836)	1075.4 (115,022)
	High-pass	56.8 (5873)	238.2 (24,858)	782.1 (83,313)	1186.3 (126,992)
**Adults**	ActiGraph	19.1 (632)	76.9 (2570)	208.1 (7164)	300.2 (10,494)
	4 Hz	30.8 (3113)	133.7 (13,692)	498.1 (52,709)	881.2 (94,188)
	10 Hz	38.9 (3983)	167.2 (17,284)	582.3 (61,748)	994.1 (106,307)
	High-pass	39.3 (4026)	170.0 (17,595)	606.2 (64,329)	1050.4 (112,371)

METs, metabolic equivalents; LPA, light physical activity; MPA, Moderate physical activity; VPA, vigorous physical activity; VVPA, very vigorous physical activity.

**Table 2 sensors-19-02186-t002:** Distribution of mean time spent at different intensities between 06.00 and 23.00.

Age Group	Filter	SED	LPA	MPA	VPA	VVPA
**Children**	ActiGraph	68%	11%	13%	3.5%	4.0%
	4 Hz	69%	16%	13%	1.2%	0.49%
	10 Hz	71%	16%	12%	0.91%	0.44%
	High-pass	73%	16%	9.8%	0.75%	0.37%
**Adults**	ActiGraph	73%	15%	12%	0.53%	0.12%
	4 Hz	72%	17%	11%	0.29%	0.072%
	10 Hz	73%	18%	9.4%	0.26%	0.096%
	High-pass	74%	16%	9.0%	0.25%	0.092%

SED, sedentary; LPA, light physical activity; MPA, Moderate physical activity; VPA, vigorous physical activity; VVPA, very vigorous physical activity.

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
