# Peer review of "Effects of Frequency Filtering on Intensity and Noise in Accelerometer-Based Physical Activity Measurements"

_sensors, 2019, doi:10.3390/s19092186_

Round 1
Reviewer 1 Report
This is a good methodology paper, which reports some interesting findings and it provides both a technological and theoretical contribution to the literature. In general, the manuscript is well-written.
Comments:
The title could focus more on the key finding.
The hypothesis about the expected outcome in the study design could be more clearly linked with the research aim.
Could the authors clarify what they mean by sedentary behaviour in the free-living setting, since it appears that there was a contribution of acceleration during that phase, which they argue could indicate noise. Moreover, why was it anticipated that noise would have a large effect at sedentary level?
Please clarify whether the subjects (adults and children) were instructed to engage in or refrain from strenuous activities (VVPA) in general in the free-living protocol. What were then the VVPA that adults and children
Table 1: SED, sedentary is not reported in the table.
Line 201: Would “dependant” be more appropriate that “depending”?
Thank you.
Author Response
Point 1: The title could focus more on the key finding.
Response 1: We agree. The title has been revised in order to focus more specifically at the aim.
Point 2: The hypothesis about the expected outcome in the study design could be more clearly linked with the research aim.
Response 2: Thank you for the comment. Our hypothesis have been revised to clearly state its relation to the aim (page 2, line 69).
Point 3: Could the authors clarify what they mean by sedentary behaviour in the free-living setting, since it appears that there was a contribution of acceleration during that phase, which they argue could indicate noise. Moreover, why was it anticipated that noise would have a large effect at sedentary level?
Response 3: Thank you for pointing out that this has to be clarified. We have previously shown that a wider filter improves the measurement of high intensity PA (Fridolfsson 2018). Our hypothesis was that since a wider filter would allow more acceleration to pass this might capture more noise. The relative contribution of noise during physical activity was expected to be negligible whereas the relative contribution would be much higher during sedentary activities. Wider filters might therefore classify more samples as light intensity instead of sedentary compared to the AG filter. The wider filters did classify some samples as LPA instead of SED but also some samples as SED instead of LPA. Some classification discrepancy is inevitable and the wider filters do not lead to less samples being classified as SED, rather the opposite with a slight increase of the total amount of SED. Our conclusions are that the wider filters might capture slightly more noise but this do not lead to a misclassification of physical activity. This has been clarified in the introduction and discussion in the revised manuscript.
Point 4: Please clarify whether the subjects (adults and children) were instructed to engage in or refrain from strenuous activities (VVPA) in general in the free-living protocol. What were then the VVPA that adults and children
Response 4: We agree that this must be clarified. All subjects were instructed not to change their physical activity habits during the measurement period. This has been added to the revised manuscript (page 3, line 141). However, the measurement itself might have affected the physical activity among subjects, despite the instructions.
Point 5: Table 1: SED, sedentary is not reported in the table.
Response 5: Thank you for noticing that SED was present in the description of Table 1. This was unintended since the table only presents the cut-points for the different physical activity intensities. Sedentary is defined as<1.5 METS which is reported in the table.
Point 6: Line 201: Would “dependant” be more appropriate that “depending”?
Response 6: We agree. This is changed in the revised manuscript.
Reviewer 2 Report
This paper aims to explore the effect of wider frequency filters on measurements of physical activity. The paper has potential to be further advanced and used in various contexts associated with filter and physical activity. However, this paper is not well organized and has the following problems to be solved.
1.Some grammar errors should be fixed. I would suggest the authors to have it proof-read by a native English speaker. some sentences such as “only small misclassification of sedentary into light activity.” should be replaced with quantitative statements.
2.The solution itself is very short lacking some in-depth analysis and discussion and many technical details are missed. Why chose the cut-offs at 4 Hz, 10 Hz?
3.The results may be a little interesting, but it is not convincing.
4.The background presentation is not solid.
5.The introduction could be extended and incorporates more comprehensive discussions in more detail. Some discussions and important references on physical activity and filter methods are missing and should be added in the introduction.
Metcalf K M, Baquero B I, Garcia M L C, et al. Calibration of the global physical activity questionnaire to Accelerometry measured physical activity and sedentary behavior[J]. Bmc Public Health, 2018, 18(1):412.
Serra M C, Balraj E, Disanzo B L, et al. Validating Accelerometry as a Measure of Physical Activity and Energy Expenditure in Chronic Stroke[J]. Topics in Stroke Rehabilitation, 2017, 24(1):18-23.
Liu L, Wang S, Hu B, et al. Learning Structures of Interval-Based Bayesian networks in Probabilistic Generative Model for Human Complex Activity Recognition[J]. Pattern Recognition, 2018, 81.
Author Response
Point 1: Some grammar errors should be fixed. I would suggest the authors to have it proof-read by a native English speaker. some sentences such as “only small misclassification of sedentary into light activity.” should be replaced with quantitative statements.
Response 1: Thank you for the comment; we agree that some quantitative statements should be added. The revised manuscript has been improved with regard to readability and quantitative statements have been added where suitable. In addition, the manuscript has now been proof-read by a native English speaker.
Point 2: The solution itself is very short lacking some in-depth analysis and discussion and many technical details are missed. Why chose the cut-offs at 4 Hz, 10 Hz?
Response 2: The rationale behind choosing the 4 Hz and 10 Hz cut-points should be stated more clearly, thank you for pointing it out for us. The work focuses on a particular part of the processing of accelerometer data to measure physical activity, frequency-filtering cut-offs. This is discussed in depth and explained with sufficient detail to ensure that there are no hinders for replication. The cut-offs at 4 Hz and 10 Hz were chosen based on biomechanical literature. The upper limit of normal human step frequency during locomotion is 4 Hz; therefore, the majority of the acceleration signal is included below this cut-point. Moreover, previous literature shows that there is acceleration related to physical activity up to 10 Hz. This has been clarified in the revised manuscript (page 3, line 153). (Fridolfsson 2018, Schepens 1998, Urbanek 2018)
Point 3: The results may be a little interesting, but it is not convincing.
Response 3: Since this study is the first investigating this topic in a free living setting the results are not definite but rather exploratory to some degree. However, the results are in line with previous research. The small effect of captured noise below 10 Hz was expected since potential noise is expected at higher frequencies (Strączkiewicz 2016). In addition, previous research have shown that wider filters more accurately capture high intensity physical activity (Fridolfsson 2018) and that the total amount of moderate to vigorous physical activity is lower (Buchan 2018).
Point 4: The background presentation is not solid.
Response 4: The background of the manuscript has been revised with regard to previous literature and the roll of objective measurement of physical activity in epidemiological research.
Point 5: The introduction could be extended and incorporates more comprehensive discussions in more detail. Some discussions and important references on physical activity and filter methods are missing and should be added in the introduction.
Metcalf K M, Baquero B I, Garcia M L C, et al. Calibration of the global physical activity questionnaire to Accelerometry measured physical activity and sedentary behavior[J]. Bmc Public Health, 2018, 18(1):412.
Serra M C, Balraj E, Disanzo B L, et al. Validating Accelerometry as a Measure of Physical Activity and Energy Expenditure in Chronic Stroke[J]. Topics in Stroke Rehabilitation, 2017, 24(1):18-23.
Liu L, Wang S, Hu B, et al. Learning Structures of Interval-Based Bayesian networks in Probabilistic Generative Model for Human Complex Activity Recognition[J]. Pattern Recognition, 2018, 81.
Response 5: Thank you for the comment. The introduction has been extended to include more details about the importance of physical activity measurements, the current methodology, its limitations and potential improvements.
Reviewer 3 Report
This paper presents the effect of wider frequency filters on measurements of physical activity, sedentary behavior and capturing of noise. Free-living accelerometer data from children and adults was processed using the different filters and intensity cut-points. The present study suggests an optimal low-pass frequency filter with a cut-off at 10 Hz to include all acceleration relevant to PA with minimal effect of noise. This technology has vast implications for physical activity measurement if properly validated.
The work is interesting and I am keen to see it published; however, there are a few changes that could be made to make the results more accessible and clear to readers, in details:
1. Maybe there lacks detailed explanations about the key contributions. The reader need more help to understand what is important, what is new, and how it relates to the state of art.
2. Figures of the experimental scene are helpful to better illustrate the laboratory facilities.
3. More comparison with the literature may be provided in the experimental results section. Some examples are:
- Vallati, C., Virdis, A., Gesi, M., Carbonaro, N., & Tognetti, A. ePhysio: A Wearables-Enabled Platform for the Remote Management of Musculoskeletal Diseases. Sensors, 2018, 19(2), 1-18.
- Albert, M. V., Azeze, Y., Courtois, M., & Jayaraman, A. In-lab versus at-home activity recognition in ambulatory subjects with incomplete spinal cord injury. Journal of NeuroEngineering and Rehabilitation, 2017, 14(1), 1–6.
- Wang, Z., Li, J., Wang, J., Zhao, H., Qiu, S., Yang, N., & Shi, X. Inertial Sensor-Based Analysis of Equestrian Sports Between Beginner and Professional Riders Under Different Horse Gaits. IEEE Transactions on Instrumentation and Measurement, 2018, 67(11), 2692–2704.
- Qiu, S., Wang, Z., Zhao, H., & Hu, H. Using Distributed Wearable Sensors to Measure and Evaluate Human Lower Limb Motions. IEEE Transactions on Instrumentation and Measurement, 2016, 65(4), 939–950.
4. What are the implications of the findings? More discussion should be provided in the manuscript. It is not enough to have a paper accepted by just reporting what they have done.
5. Other minor concerns:
-How do you deal with sensor misplacement?
-Proofread the paper and improve readability.
Author Response
Point 1: Maybe there lacks detailed explanations about the key contributions. The reader need more help to understand what is important, what is new, and how it relates to the state of art.
Response 1: Thank you for pointing out how to help the reader to understand the contributions better. The introduction have been extended to give a better background and emphasize how this study aim to improve the deficits in the most commonly used methodology. The discussion is revised in order to explain more clearly how the results of the current study could improve this methodology and what implications it could have for research about the relation between physical activity and health.
Point 2: Figures of the experimental scene are helpful to better illustrate the laboratory facilities.
Response 2: Good point. Figures are available in the article referred to regarding the lab data; this has been clarified in the revised manuscript (page 3, line 135). (Fridolfsson 2018)
Point 3: More comparison with the literature may be provided in the experimental results section. Some examples are:
- Vallati, C., Virdis, A., Gesi, M., Carbonaro, N., & Tognetti, A. ePhysio: A Wearables-Enabled Platform for the Remote Management of Musculoskeletal Diseases. Sensors, 2018, 19(2), 1-18.
- Albert, M. V., Azeze, Y., Courtois, M., & Jayaraman, A. In-lab versus at-home activity recognition in ambulatory subjects with incomplete spinal cord injury. Journal of NeuroEngineering and Rehabilitation, 2017, 14(1), 1–6.
- Wang, Z., Li, J., Wang, J., Zhao, H., Qiu, S., Yang, N., & Shi, X. Inertial Sensor-Based Analysis of Equestrian Sports Between Beginner and Professional Riders Under Different Horse Gaits. IEEE Transactions on Instrumentation and Measurement, 2018, 67(11), 2692–2704.
- Qiu, S., Wang, Z., Zhao, H., & Hu, H. Using Distributed Wearable Sensors to Measure and Evaluate Human Lower Limb Motions. IEEE Transactions on Instrumentation and Measurement, 2016, 65(4), 939–950.
Response 3: Thank you for the comment. Since there are only a few studies about the effect of different frequency filtering of physical activity data, all available studies that can be used for direct comparisons are being discussed. The suggested literature is not specifically addressing the issues investigated in the current manuscript. However, the discussion is revised in order to put the results in a context and highlight the implications.
Point 4: What are the implications of the findings? More discussion should be provided in the manuscript. It is not enough to have a paper accepted by just reporting what they have done.
Response 4: Thank you for the comment, this is important and should be made clearer. The discussion is revised in order to better explain how the results relate to previous literature and what the implications might be for the relationship between physical activity and health.
Point 5: Other minor concerns:
-How do you deal with sensor misplacement?
-Proofread the paper and improve readability.
Response 5a: The sensors are usually worn in an elastic belt provided. The direction of the sensor is not important if the vector magnitude is considered and previous research has shown that the left hip yields similar results as the right (Aadland 2015).
5b: The revised manuscript has been improved with regard to readability, and has been proof-read by an native English speaker.
Round 2
Reviewer 2 Report
1. Some expressions are still out of place, such as “It is known, that” in line 57. The expressions such as “ActiGraph (AG) is the most commonly used accelerometer” and “counts is still the most used metric” are not accurate.
2. What kind of data do you use? Raw data or counts data? It seems that you choose 4hz and 10hz because you just take the step frequency into account, filtering raw data or filtering counts data? Some details of the solution are not clear.
3. “previous studies have shown that acceleration related to physical activity is found at frequencies up to 10 Hz”,The results shows that “an optimal low-pass frequency filter with a cut-off at 10 Hz to include all acceleration relevant to PA with minimal effect of noise”. The result is not convincing and only shows it is an optimal low-pass frequency among 4hz, 10hz and removed.
4. It will be more convincing if you compared this wok with other people's previous research.
5. The background presentation is not solid. The introduction could be extended and incorporates more comprehensive discussions in more detail. Some discussions and important references on physical activity are missing and should be added in the introduction.
Metcalf K M, Baquero B I, Garcia M L C, et al. Calibration of the global physical activity questionnaire to Accelerometry measured physical activity and sedentary behavior[J]. Bmc Public Health, 2018, 18(1):412.
Serra M C, Balraj E, Disanzo B L, et al. Validating Accelerometry as a Measure of Physical Activity and Energy Expenditure in Chronic Stroke[J]. Topics in Stroke Rehabilitation, 2017, 24(1):18-23.
Liu L, Wang S, Hu B, et al. Learning Structures of Interval-Based Bayesian networks in Probabilistic Generative Model for Human Complex Activity Recognition[J]. Pattern Recognition, 2018, 81.
Author Response
Comment 1. Some expressions are still out of place, such as “It is known, that” in line 57. The expressions such as “ActiGraph (AG) is the most commonly used accelerometer” and “counts is still the most used metric” are not accurate.
Response 1. Thank you for addressing these specific issues.
a) “It is known, that the step frequency is higher in children than adults and during more intensive PA in both adults and children [12,13]” (page 2, line 61)
This is a very well established fact in biomechanical literature and the statement is followed by suitable references.
b) “ActiGraph (AG) is the most commonly used accelerometer” and “counts is still the most used metric”
Our statements regarding this are accurate and we stick to that ActiGraph is the most commonly used accelerometer in physical activity measurement (Wijndaele 2015), and counts is still the most used metric (Migueles 2017). However, this has been clarified in the text of the manuscript with added references (page 1-2, line 44-45). We acknowledge that in some fields of research other sensors are more common and that other sensors are used more and more in epidemiological research.
Comment 2. What kind of data do you use? Raw data or counts data? It seems that you choose 4hz and 10hz because you just take the step frequency into account, filtering raw data or filtering counts data? Some details of the solution are not clear.
Response 2. Thank you for the question. We extract the raw data and apply different filters on the raw data to generate the output. The processing is the same as with the original ActiGraph counts, but apart from the standard ActiGraph filter at 1.63 Hz, we instead apply different filters and present the data as mean mg instead of aggregated 8-bit resolution counts. Figure 1 (page 4, line 163) states that we apply the filters on 30 Hz raw data and explains the rest of the processing of the raw data. Further details are explained in the text of the methods section.
The 4 Hz and 10 Hz cut-offs are not primarily based on the step frequency but instead the frequency spectrum of acceleration related to physical activity (Fridolfsson 2018, Strączkiewicz 2016).
Comment 3. “previous studies have shown that acceleration related to physical activity is found at frequencies up to 10 Hz”, The results shows that “an optimal low-pass frequency filter with a cut-off at 10 Hz to include all acceleration relevant to PA with minimal effect of noise”. The result is not convincing and only shows it is an optimal low-pass frequency among 4hz, 10hz and removed.
Response 3. Thank you for letting us clarify this. Just as you refer to, previous studies have suggested that 10 Hz filtering might be an optimal low pass cut-off for capturing physical activity.
a) Acceleration related to physical activity is found up to this cut-off. (Fridolfsson 2018, Urbanek 2018)
b) When moving at the same locomotion speed, 10 Hz filtered acceleration displays the least within age-group variation. Which is explained biomechanically by more accurate capturing of individual gait patterns. (Fridolfsson 2018)
c) Car driving adds noise to accelerometer measurements in the frequency span from 10 Hz and upwards depending on speed. (Strączkiewicz 2016)
Since this is the first study that investigates different filters vulnerability to capturing noise, it is somewhat exploratory. However, we do state that the study suggests that 10 Hz might be an optimal cut-off. This is based on the results of our study showing that neither 4 Hz, 10 Hz nor unfiltered acceleration seems to be particularly vulnerable to capturing noise in a free-living setting. This alone is not sufficient to claim that 10 Hz might be an optimal filter, but together with previous research, the results are convincing. Previous research show that 10 Hz might be an optimal cut-off from a biomechanical point of view and that including acceleration above 10 Hz might capture noise from specific activities such as car driving.
Comment 4. It will be more convincing if you compared this work with other people's previous research.
Response 4. We agree that the relation of the results from the current study to other people’s previous research is of great importance for its credibility. However, since there are very few studies investigating this specific topic, we believe that all relevant literature is being discussed.
Comment 5. The background presentation is not solid. The introduction could be extended and incorporates more comprehensive discussions in more detail. Some discussions and important references on physical activity are missing and should be added in the introduction.
Metcalf K M, Baquero B I, Garcia M L C, et al. Calibration of the global physical activity questionnaire to Accelerometry measured physical activity and sedentary behavior[J]. Bmc Public Health, 2018, 18(1):412.
Serra M C, Balraj E, Disanzo B L, et al. Validating Accelerometry as a Measure of Physical Activity and Energy Expenditure in Chronic Stroke[J]. Topics in Stroke Rehabilitation, 2017, 24(1):18-23.
Liu L, Wang S, Hu B, et al. Learning Structures of Interval-Based Bayesian networks in Probabilistic Generative Model for Human Complex Activity Recognition[J]. Pattern Recognition, 2018, 81.
Response 5. Thank you for the comment. The goal of this study together with our previous studies (Brønd 2017, Fridolfsson 2018) is to target a specific problem that has affected physical activity measurement in epidemiology negatively in a broad sense, namely the consequence of using the ActiGraph counts to assess activity intensity. Its measurement error has influenced our understanding of physical activity measurement, as it is the most common measure the latest decades. We have used a biomechanical model to understand the measurement error and to develop an improved measure (Fridolfsson 2018). Other solutions have recently been offered (e.g. MAD, ENMO, machine-learning models) but have not provided the understanding what we actually capture with accelerometers at different placements when we try to assess activity intensity. It seems that simple linear models provide the same accuracy to asses activity intensity for the hip placement as more advanced machine-learning approaches, while machine-learning would be required for the wrist placement (Montoye 2017). It also seems that acceleration data collected at the hip and filtered with a 10Hz low-pass filter follows biomechanical theory of mechanical work (Fridolfsson 2018).
To stay focused on our goal to achieve an improved measure of activity intensity solving the measurement error of previous method, we believe that the background is sufficient for introducing the reader to the problem targeted. A more comprehensive review in the background section would of course provide the reader with a better understanding of the whole field, but this is not the scope of this paper and would make our manuscript to get out of focus and become too large. In line with this, we believe the suggested literature does not address the specific issues investigated in the current study. We hope the results from our studies would contribute to all researchers that have relied on the ActiGraph monitor to get a simple measure of activity intensity to be used in epidemiology, but also to those using other accelerometers providing raw acceleration data to assess activity intensity.
References
Fridolfsson, J.; Börjesson, M.; Arvidsson, D.; Fridolfsson, J.; Börjesson, M.; Arvidsson, D. A Biomechanical Re-Examination of Physical Activity Measurement with Accelerometers. Sensors 2018, 18, 3399.
Migueles, J.H.; Cadenas-Sanchez, C.; Ekelund, U.; Nyström, C.D.; Mora-Gonzalez, J.; Löf, M.; Labayen, I.; Ruiz, J.R.; Ortega, F.B. Accelerometer Data Collection and Processing Criteria to Assess Physical Activity and Other Outcomes: A Systematic Review and Practical Considerations. Sports Med. 2017, 47, 1821–1845.
Montoye, A.H.K.; Begum, M.; Henning, Z.; Pfeiffer, K.A. Comparison of linear and non-linear models for predicting energy expenditure. Physiol. Meas. 2017, 38, 343–357.
Strączkiewicz, M.; Urbanek, J.; Fadel, W.; Crainiceanu, C.; Harezlak, J. Automatic car driving detection using raw accelerometry data. Physiol. Meas. 2016, 37, 1757–1769.
Urbanek, J.K.; Zipunnikov, V.; Harris, T.; Fadel, W.; Glynn, N.; Koster, A.; Caserotti, P.; Crainiceanu, C.; Harezlak, J. Prediction of sustained harmonic walking in the free-living environment using raw accelerometry data. Physiol. Meas. 2018, 39, 02NT02.
Wijndaele, K.;Westgate, K.; Stephens, S.K.; Blair, S.N.; Bull, F.C.; Chastin, S.F.; Dunstan, D.W.; Ekelund, U.; Esliger, D.W.; Freedson, P.S.; et al. Utilization and harmonization of adult accelerometry data: Review and expert consensus. Med. Sci. Sports Exerc. 2015, 47, 2129.